# REINFORCED GENETIC ALGORITHM LEARNING FOR OPTIMIZING COMPUTATION GRAPHS

**Aditya Paliwal** *
Google Research
adipal@google.com

**Felix Gimeno**
DeepMind
fgimeno@google.com

**Vinod Nair**
DeepMind
vinair@google.com

**Yujia Li**
DeepMind
yujiali@google.com

**Miles Lubin**
Google Research
mlubin@google.com

**Pushmeet Kohli**
DeepMind
pushmeet@google.com

**Oriol Vinyals**
DeepMind
vinyals@google.com

## ABSTRACT

We present a deep reinforcement learning approach to minimizing the execution cost of neural network computation graphs in an optimizing compiler. Unlike earlier learning-based works that require training the optimizer on the same graph to be optimized, we propose a learning approach that trains an optimizer offline and then generalizes to previously unseen graphs without further training. This allows our approach to produce high-quality execution decisions on real-world TensorFlow graphs in seconds instead of hours. We consider two optimization tasks for computation graphs: minimizing running time and peak memory usage. In comparison to an extensive set of baselines, our approach achieves significant improvements over classical and other learning-based methods on these two tasks.

## 1 INTRODUCTION

Deep Learning frameworks such as MXNet (Chen et al., 2015), PyTorch (Paszke et al., 2017), and TensorFlow (TensorFlow Authors, 2016a) represent neural network models as computation graphs. Efficiently executing such graphs requires optimizing discrete decisions about how to map the computations in a graph onto hardware so as to minimize a relevant cost metric (e.g., running time, peak memory). Given that execution efficiency is critical for the success of neural networks, there is growing interest in the use of optimizing static compilers for neural network computation graphs, such as Glow (Rotem et al., 2018), MLIR (MLIR Authors, 2018), TVM (Chen et al., 2018a), and XLA (XLA team, 2017).

Here we consider the *model parallelism* setting where a computation graph can be executed using multiple devices in parallel. Nodes of the graph are computational tasks, and directed edges denote dependencies between them. We consider *jointly optimizing* over *placement*, i.e., which nodes are executed on which devices, and *schedule*, i.e., the node execution order on each device. These decisions are typically made in either one or two passes in the compiler. We consider two different objectives: 1) minimize running time, subject to not exceeding device memory limits, and 2) minimize peak memory usage. In the optimization literature, such problems are studied under the class of *task scheduling*, which is known to be NP-hard in typical settings (Sinnen, 2007; Kwok & Ahmad, 1999).

As scheduling and placement are just a few of the many complex decisions made in a compiler, it is essential in a production setting that a solution 1) produce solutions of acceptable quality fast, even on large graphs (e.g., thousands of nodes) and decision spaces, and 2) handle diverse graphs from various types of applications, neural network architectures, and users. In this work we consider

---

*Google AI Resident

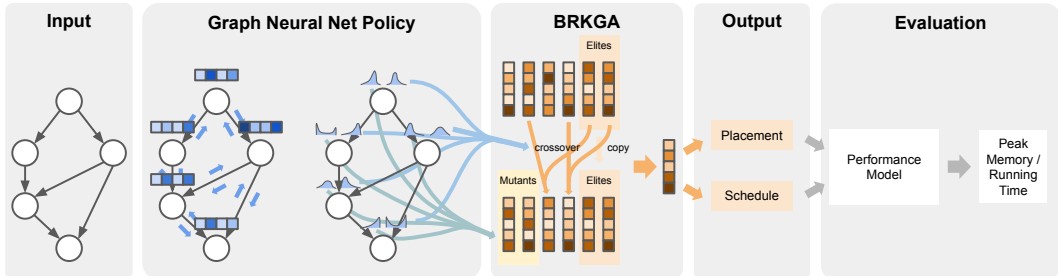

Fig. 1: Overview of our approach. The Biased Random Key Genetic Algorithm (BRKGA) is used to optimize execution decisions for a computation graph (e.g., placement and scheduling of nodes) with respect to a cost metric (e.g., running time, peak memory) computed using the performance model. BRKGA requires proposal distributions for each node in the graph to generate candidate solutions in its search loop. The default choice is agnostic to the input graph: uniform distribution over $[0, 1]$ at all nodes. We use a graph neural network policy to predict node-specific non-uniform proposal distribution choices (parameterized as beta distributions over $[0, 1]$). BRKGA is then run with those choices and outputs the best solution found by its iteration limit. By controlling the non-uniformity of the distributions, the policy directs how BRKGA's search effort is allocated such that a better solution can be found with the same search budget.

*learning an optimizer* that satisfies these requirements. Crucially, we aim to learn an optimizer that generalizes to a broad set of previously unseen computation graphs, without the need for training on such graphs, thus allowing it to be fast at test time.

Previous works on learning to optimize model parallelism decisions (Mirhoseini et al., 2017; 2018; Addanki et al., 2019) have not considered generalization to a broad set of graphs nor joint optimization of placement and scheduling. In Mirhoseini et al. (2017; 2018), learning is done from scratch for each computation graph and for placement decisions only, requiring hours (e.g., 12 to 27 hours per graph). This is too slow to be broadly useful in a general-purpose production compiler. We propose an approach that takes only seconds to optimize similar graphs. In concurrent work to ours, Addanki et al. (2019) shows generalization to unseen graphs, but they are generated artificially by architecture search for a single learning task and dataset. In contrast, we collect real user-defined graphs spanning a broad set of tasks, architectures, and datasets. In addition, both Mirhoseini et al. (2017; 2018) and Addanki et al. (2019) consider only placement decisions and rely on TensorFlow's dynamic scheduler; they do not address the static compiler setting where it is natural to jointly optimize scheduling and placement.

The key idea of our approach (Figure 1) is to learn a neural network that, conditioned on the input graph to be optimized, directs an existing optimization algorithm's search such that it finds a better solution in the same search budget. We choose the Biased Random-Key Genetic Algorithm (BRKGA (Gonçalves & Resende, 2011)) as the optimization algorithm after an extensive evaluation of several choices showed that it gives by far the best speed-vs-quality trade-off for our application. BRKGA produces good solutions in just a few seconds even for real-world TensorFlow graphs with thousands of nodes, and we use learning to improve the solution quality significantly at similar speed. We train a graph neural network (Battaglia et al., 2018) to take a computation graph as input and output node-specific proposal distributions to use in the mutant generation step of BRKGA's inner loop. BRKGA is then run to completion with those input-dependent distribution choices, instead of input-agnostic default choices, to compute execution decisions. The distributions are predicted at each node, resulting in a high-dimensional prediction problem. There is no explicit supervision available, so we use the objective value as a reward signal in a contextual bandit approach with REINFORCE (Williams, 1992). Our approach, "Reinforced Genetic Algorithm Learning" (REGAL), uses the network's ability to generalize to new graphs to significantly improve the solution quality of the genetic algorithm for the same objective evaluation budget.

We follow the static compiler approach of constructing a coarse static cost model to evaluate execution decisions and optimizing them with respect to it, as done in (Addanki et al., 2018; Jia et al., 2018). This is in contrast to evaluating the cost by executing the computation graph on hardware (Mirhoseini

et al., 2017; 2018). A computationally cheap cost model enables fast optimization. It is also better suited for distributed training of RL policies since a cost model is cheap to replicate in parallel actors, while hardware environments are not. Our cost model corresponds to classical NP-hard scheduling problems, so optimizing it is difficult. In this paper we focus fully on learning to optimize this cost model, leaving integration with a compiler for future work.

We structure the neural network's task as predicting proposal distributions to use in the search over execution decisions, rather than the decisions themselves directly. Empirically we have found the direct prediction approach to be too slow at inference time for our application and generalizes poorly. Our approach potentially allows the network to learn a more abstract policy not directly tied to detailed decisions that are specific to particular graphs, which may generalize better to new graphs. It can also make the learning task easier as the search may succeed even with sub-optimal proposal distribution predictions, thus smoothening the reward function and allowing the network to incrementally learn better proposals. The node-specific proposal distribution choices provide a rich set of knobs for the network to flexibly direct the search. Combining learning with a search algorithm has been shown to be successful (e.g., (Silver et al., 2017; 2018)), and our work can be seen as an instance of the same high-level idea.

This paper makes several contributions:

- We are the first to demonstrate learning a policy for jointly optimizing placement and scheduling that generalizes to a broad set of real-world TensorFlow graphs. REGAL significantly outperforms all baseline algorithms on two separate tasks of minimizing runtime and peak memory usage (section 5.3) on datasets constructed from 372 unique real-world TensorFlow graphs, the largest dataset of its kind in the literature and at least an order of magnitude larger than the ones in previous works (Mirhoseini et al., 2017; 2018; Chen et al., 2018b; Addanki et al., 2018; 2019).

- We use a graph neural network to predict mutant sampling distributions of a genetic algorithm, specifically BRKGA, for the input graph to be optimized. This directs BRKGA's search in an input-dependent way, improving solution quality for the same search budget.

- We compare extensively to classical optimization algorithms, such as enumerative search, local search, genetic search, and other heuristics, and analyze room-for-improvement in the objective value available to be captured via learning. Both are missing in previous works.

## 2    RELATED WORK

*Learning to optimize computation graphs:* AutoTVM (Chen et al., 2018b) applies learning to the very different problem of optimizing low-level implementations of operators in a tensor program, while we focus on optimizing higher-level decisions such as placement and scheduling of ops. Mao et al. (2019) use graph neural nets and RL to learn a scheduling policy for data processing jobs on clusters. These works are conceptually similar to ours in their use of learning, applied to a different domain.

*Learning for combinatorial optimization:* Our work is an instance of applying learning for combinatorial optimization (Bengio et al., 2018). Previous works on learning graph combinatorial optimization algorithms (e.g., Li et al. (2018); Khalil et al. (2017)) have focused on problems such as Minimum Vertex Cover, Maximum Clique, Maximum Independent Set, etc. The task scheduling problem we consider is significantly different in that the objective value is a more complex function on node-level decisions. Also, we focus on large-scale, real-world TensorFlow graphs, while e.g., Khalil et al. (2017) uses small-scale, synthetic graph distributions.

*Learning a proposal distribution for stochastic search:* Bunel et al. (2017) learns a policy for predicting instance-dependent proposal distributions to be used in the stochastic optimizer STOKE (Schkufza et al., 2013) for superoptimizing programs. However, it uses handcrafted instance features and shows results on relatively simple, small programs. In contrast, we automatically learn the instance representations and show results on real-world graphs. An earlier work by Paige & Wood (2016) similarly learns a neural network to predict input-dependent proposal distributions for sequential Monte Carlo search for inference in a graphical model.

*Optimization without learning:* Parallel task scheduling (Sinnen, 2007; Kwok & Ahmad, 1999) is a classical problem for scheduling ops in a computational graph to minimize runtime. Learning is not

traditionally a part of the approaches proposed in this literature. Mayer et al. (2017) studies greedy task scheduling approaches for TensorFlow. Jia et al. (2018) develops a simulation-based optimizer for deep learning computation graphs that uses a larger decision space by combining data, model, and attribute parallelism. Our approach can potentially be extended to such larger decisions spaces to achieve even bigger improvements in execution cost.

## 3 BACKGROUND

Figure 1 shows an overview of our approach. Given an input graph to optimize, instead of applying BRKGA directly with the default uniform distribution at all nodes, a graph neural network predicts beta distribution choices at each node. BRKGA is run with these choices to optimize placement and scheduling decisions with respect to the objective defined by the performance model. We first explain the performance model and BRKGA in this section, and the learning component in the next.

### 3.1 PERFORMANCE MODEL

A computation graph has a set of ops to run. Each op produces zero or more tensors and requires zero or more tensors as input. The runtime of each op is known and fixed (e.g., given by a simulator as in Jia et al. (2018)). The memory use of each tensor is known (an assumption that holds in static compilers like XLA). We assume a collection of $d$ homogeneous devices that have separate local memory and can run at most one op at a time. An op can run only when its input tensors are present in the local memory. Tensors can be transferred across devices by synchronous (blocking) transfers. Tensors are freed from local memory after all local consumers have run.

In this setting, we consider the problem of finding an assignment of ops to devices and an overall schedule such that each op is run once with the objectives of (1) minimizing the peak local memory use across devices (e.g., to find a feasible way to run a large computation graph), or (2) minimizing the runtime subject to a constraint on the peak memory used on any device.

The performance model does not consider rematerialization of tensors, fragmentation when computing memory use, and asynchronous transfers between devices. Despite these simplifications, the model yields slight variants of problems that are known to be NP-hard (Eyraud-Dubois et al., 2015) and therefore remains a challenging setting in which to study how to learn an optimizer. See section A.4 for more details of the model.

### 3.2 BIASED RANDOM-KEY GENETIC ALGORITHM

Biased random-key genetic algorithm (BRKGA) is a meta-heuristic framework that has been successful in a wide array of applications for solving hard combinatorial optimization problems (Gonçalves & Resende, 2011). In BRKGA, *chromosomes* in a population are encoded as $n$-dimensional vectors with entries in $[0, 1]$ for some fixed $n$. This *random-key* encoding decouples the application from the genetic algorithm, specifically the crossover and mutant generation procedures (Bean, 1994).

The BRKGA variant we use is specified by (1) a fitness evaluation function $f : [0, 1]^n \to \mathbb{R}$, (2) scalar integer parameters $\pi$, $\pi_e$, and $\pi_c$ representing the population size, number of elites, and number of children, resp., (3) an elite bias $\rho \in [0.5, 1.0)$, and (4) a mutant generation distribution $\mathcal{D}$ over $[0, 1]^n$. The procedure aims to find a chromosome that maximizes $f$.

The initial population (a collection of $\pi$ chromosomes) is created by sampling from $\mathcal{D}$. (Known good solutions may also be used to initialize a population.) One evolution step is completed as follows.

1. Sort the chromosomes in order of decreasing fitness using $f$. Denote the first $\pi_e$ chromosomes as *elites* and the remaining chromosomes as *nonelites*.

2. Construct the next generation from three different sources of chromosomes: (a) Copy the elite chromosomes unmodified from the last generation. (b) For each of the $\pi_c$ new children, select two parent chromosomes uniformly at random, one from the nonelites and one from the elites. Apply the crossover procedure (described below) to generate a new chromosome given the two parents. (c) Generate the remaining $\pi - \pi_e - \pi_c$ by sampling from $\mathcal{D}$.

We continue the evolution procedure for a fixed number of evaluations, i.e., calls to $f$. Given an elite and nonelite chromosome $\boldsymbol{a}, \boldsymbol{b} \in [0, 1]^n$ (resp.), the crossover procedure produces a child chromosome $\boldsymbol{c}$ by independently combining entries from the parents. Specifically, for each index $i \in 1, \ldots, n$ independently, let $\boldsymbol{c}_i = \boldsymbol{a}_i$ with probability $\rho$ and $\boldsymbol{c}_i = \boldsymbol{b}_i$ with probability $1 - \rho$.

Our use of BRKGA is standard except for the mutant-sampling distribution $\mathcal{D}$, which is usually fixed to the uniform distribution. We generalize BRKGA for instance-specific learning by sampling from $n$ independent beta distributions, whose parameters can vary by index. Beta flexibly admits non-uniform distribution choices and also subsumes the uniform choice.

Given a computation graph, let $d$ be the number of devices, $o$ the number of ops, and $t$ the number of tensors. We define the chromosome encoding a scheduling solution to have three distinct parts: (1) $o \times d$ entries specifying op-to-device affinities; (2) $o$ entries specifying scheduling priorities for each op; (3) $t \times d$ entries specifying tensor-to-device priorities for transfers that may be needed. Given a chromosome, op placements are picked by maximum affinity. Transfer ops are created as implied by the placements. We then obtain a schedule by performing a topological sort over the ops given their tensor dependencies, breaking ties by using the corresponding node priorities. Once a schedule is constructed, the performance model is used to evaluate its peak memory and/or runtime. When enforcing a memory constraint, the fitness of a schedule is encoded such that all memory-feasible schedules have better fitness than infeasible schedules. An example is provided in section A.5.

## 4 REGAL

### 4.1 CONTEXTUAL BANDIT FORMULATION

We train a contextual bandit policy that predicts beta distribution choices for each of the nodes of a computation graph to be used by BRKGA to optimize it. For each round in the bandit setting, first the *context* is observed by drawing a computation graph $G$ as an i.i.d. sample from a distribution $\mathcal{D}$ (e.g., a distribution over TensorFlow graphs). $G$ has a set of nodes $V$ and a set of edges $E$, with features associated with the nodes and edges. A *policy* $p(\boldsymbol{a}|G)$ is applied to make a set of decisions at each node. These decisions, denoted $\boldsymbol{a}_v$ for each $v \in V$, across all nodes form one *action* $\boldsymbol{a} = \{\boldsymbol{a}_{v \in V}\}$. One decision in $\boldsymbol{a}$ corresponds to playing one arm of the bandit, and specifying the entire $\boldsymbol{a}$ corresponds to playing several arms together in a single round. This can be viewed as a combinatorial multi-armed bandit problem (Chen et al., 2013).

The action $\boldsymbol{a}$ specifies all the node-specific beta distributions BRKGA needs to optimize placement and scheduling decisions for $G$. To enable a policy over discrete choices, we quantize the mean and variance parameters of the beta distribution. The environment then runs BRKGA with those distribution choices with a fixed iteration limit. The final objective value is used to compute the reward. To make the reward values comparable across different graphs, we divide the objective value $o_a(G)$ achieved on a graph $G$ with action $\boldsymbol{a}$ by the objective value $o_s(G)$ achieved by standard BRKGA using uniform distributions. Since we want to minimize the objective (e.g., runtime or peak memory), we define the reward as $r(\boldsymbol{a}, G) = -\frac{o_a(G)}{o_s(G)}$. So a reward $> -1$ corresponds to an action that achieves a better objective value than standard BRKGA on a graph.

We maximize the expected reward $L = \mathbb{E}_G\left[\sum_{\boldsymbol{a}} p(\boldsymbol{a}|G)r(\boldsymbol{a}, G)\right]$, where $\mathbb{E}_G$ is an expectation over graphs in our training set. Learning is done by REINFORCE (Williams, 1992). We added a scalar baseline $b(G)$ to reduce the variance of the gradient estimates.

### 4.2 GRAPH NEURAL NETWORK POLICY

From computation graphs, we derive multigraphs with attributed nodes and directed edges. Denote a multigraph $G = (V, E)$. In our setup, the nodes $V$ correspond 1:1 to the ops. An edge $e \in E$ exists from $u$ to $v$ for each tensor that op $v$ requires that is produced by op $u$. As a tensor can be required by multiple ops, the correspondence from edges to tensors may be many to one. Each node $v \in V$ and edge $e \in E$ has an attribute vector $\boldsymbol{x}_v$ and $\boldsymbol{x}_e$. The attributes contain respective features, e.g., sizes of the tensors.

We learn a model that predicts good mutant sampling distributions for BRKGA given this multigraph. Each node has $d + 1$ independent beta distributions, corresponding to device affinities and scheduling

priorities, whose parameters are represented as a vector $\boldsymbol{a}_v$. These are the model's actions in RL terms, and our model specifies a distribution over actions $\boldsymbol{a} = \{\boldsymbol{a}_v\}_{v \in V}$ for each graph, $p(\boldsymbol{a}|G)$. Note the action space is different from graph to graph.

We use Graph Neural Networks (GNNs) (Scarselli et al., 2009; Li et al., 2015; Gilmer et al., 2017; Battaglia et al., 2018) to learn representations for computation graphs. Given a (multi)graph $G$, a GNN computes representation vectors $\boldsymbol{h}_v$ for each node through an iterative message passing process as follows:

$$\begin{aligned}
\boldsymbol{h}_v^{(0)} &= \mathrm{MLP}_n(\boldsymbol{x}_v), & \boldsymbol{h}_e &= \mathrm{MLP}_e(\boldsymbol{x}_e) \\
\boldsymbol{m}_e^{(t)} &= \mathrm{MLP}_{\mathrm{msg}}([\boldsymbol{h}_{e_s}^{(t)}, \boldsymbol{h}_{e_t}^{(t)}, \boldsymbol{h}_e]), & \boldsymbol{m}_e^{(t)'} &= \mathrm{MLP}_{\mathrm{msg}}'([\boldsymbol{h}_{e_s}^{(t)}, \boldsymbol{h}_{e_t}^{(t)}, \boldsymbol{h}_e]) \\
\boldsymbol{h}_v^{(t+1)} &= \mathrm{MLP}_{\mathrm{node}}\left(\left[\boldsymbol{h}_v^{(t)}, \sum_{e:e_t=v} \boldsymbol{m}_e^{(t)} + \sum_{e:e_s=v} \boldsymbol{m}_e^{(t)'}\right]\right),
\end{aligned} \qquad (1)$$

where $e_s$ is the source node of edge $e$ and $e_t$ is the target node. In our formulation, $\mathrm{MLP}_n$ and $\mathrm{MLP}_e$ are multilayer perceptrons (MLPs) that encode node and edge attributes, $\mathrm{MLP}_{\mathrm{msg}}$ and $\mathrm{MLP}_{\mathrm{msg}}'$ compute messages along the edges in the edge direction ($\boldsymbol{m}_e^{(t)}$) and the opposite direction ($\boldsymbol{m}_e^{(t)'}$), $\mathrm{MLP}_{\mathrm{node}}$ updates node representations and $[.]$ represents flat vector concatenation. After $T$ rounds of message passing, the representation for each node $\boldsymbol{h}_v = \boldsymbol{h}_v^{(T)}$ will contain information from the $T$-hop neighborhood around $v$ in the graph. Given the $\boldsymbol{h}_v$'s, we produce $\boldsymbol{a}_v$'s through conditionally independent predictions, where the prediction for one node $v$ does not depend on the predictions of other nodes given the computed representations:

$$p(\boldsymbol{a}|G) = \prod_v p(\boldsymbol{a}_v|G) = \prod_v p(\boldsymbol{a}_v|\mathrm{MLP}_a(\boldsymbol{h}_v)). \qquad (2)$$

$\mathrm{MLP}_a$ is shared across all nodes for predicting the parameters of the output distributions. In our experiments, we quantize the continuous beta distribution parameters and use a discrete action space. The outputs are therefore categorical, and we use the MLP to compute the logits of the corresponding softmax distributions. More details are included in section A.6. The baseline is computed using a separate GNN, where after we obtained the node representations $\boldsymbol{h}_v$, we aggregate across nodes and compute $b(G)$ as $b(G) = \mathrm{MLP}_b\left(\frac{1}{|V|}\sum_v \mathrm{MLP}_g(\boldsymbol{h}_v)\right)$.

## 5 EXPERIMENTAL RESULTS

### 5.1 TASKS AND DATASETS

We consider two tasks, one is minimizing peak memory and the other is minimizing running time, both on two homogeneous devices with 16 GiB of memory each and synchronous tensor transfers with zero cost (zero latency and infinite bandwidth). We train a separate neural network for each task-dataset pair for the case of two devices.

We have collected a dataset of 372 topologically-unique real-world TensorFlow graphs by mining machine learning jobs on Google's internal compute cluster (see A.1.2). These jobs are from a wide range of production and research use cases. The dataset is split into {train, valid, test} sets containing {60%, 20%, 20%} graphs, respectively. These sets are disjoint with respect to graph topology, so at test time the policy needs to generalize to new topologies.

We augment the dataset by applying multiplicative noise to tensor sizes and op running times to create several variants per graph. Even though the variants of the same graph share the same topology, they represent different optimization problem instances. We create separate datasets for minimizing runtime and peak memory. The *TF runtime dataset* has 16329 training, 5470 validation, and 5266 test graphs. The *TF peak memory dataset* has 22400 training, 7400 validation, and 7400 test graphs.

For reproducibility, we have released[1] a synthetic dataset of computation graphs with 10000 training, 1000 validation, and 1000 test cases. The graph topologies are generated from several classical random graph models, and the op running times and tensor sizes are sampled from Gaussian distributions (see A.1.4). On this dataset we minimize running time without a memory constraint (e.g., on two homogeneous devices with infinite memory).

---

[1] https://drive.google.com/drive/folders/1lxRl1ocsWu-POwbdEY06Mrzq6Ot99j7N

## 5.2 BASELINES

*Graph Partitioning + Depth First Search (GP+DFS):* Combines a graph partitioning (GP) baseline for device placement to minimize communication across devies and a Depth-First Search heuristic similar to the one implemented in XLA (TensorFlow Authors, 2016b) to compute per-device schedules given placements. This is representative of the XLA compiler's solution for model parallelism.

*Local Search*: The method starts with a random placement and schedule and greedily improves it by moving an op across devices or changing an op's order in the current schedule.

*Graph-As-Sequence Model (GAS)*: Like Mirhoseini et al. (2017; 2018), we convert the graph into a sequence using a topological sort and apply a recurrent neural network to predict node-level distributions to be used by BRKGA. This comparison measures the usefulness of graph structure for learning.

*BRKGA XK*: Run BRKGA for $X$ thousand iterations with uniform sampling distributions using default hyperparameters consistent with Gonçalves & Resende (2011). This comparison measures the performance of the default version of BRKGA.

*Tuned BRKGA*: Apply grid search to BRKGA's hyperparameters on the training set and pick the best. This represents how well BRKGA performs by customizing it to the distribution of computation graphs, but without instance-dependent customization.

*Instance-dependent Random Search (IDRS)*: Same as REGAL, but BRKGA is replaced with random search. This is done by running BRKGA for only one generation using the proposal distributions computed by the neural network.

Additionally, we use a Constraint Programming (CP) approach with the CP-SAT solver of Google OR-tools (Google, 2019) to establish a provably global optimum for each computation graph optimization problem instance by running for up to 24 hours. As an enumerative algorithm, it is generally not competitive when run only for seconds.

For a fair comparison, we fix the number of performance model evaluations allowed per graph to be the same across algorithms. (Except GP+DFS, which does not allow fixing it.) Given typical TensorFlow graph sizes and compiler running time constraints, we estimate that a budget of 5000 evaluations is feasible in practice, so we use that in the experiments.

**Learning to directly predict a solution:** We have explored two more approaches for training a graph neural network to predict placement and scheduling solutions directly, without BRKGA. We used supervised learning to train a network to predict BRKGA's solutions. The best accuracy was achieved by predicting the solution autoregressively, one variable at a time conditioned on previously predicted variables. We also used RL to learn a policy with IMPALA (Espeholt et al., 2018) to optimize the objective value by incrementally predicting the solution one variable at a time, and once complete, iteratively improving it with a learned local search policy. The inference cost for both approaches is quadratic in the number of nodes (the graph net is applied a linear number of times, each with linear cost), while REGAL's inference cost is linear, making them orders of magnitude slower than REGAL at test time. An evaluation on a test set of small graphs showed that neither approach improves on BRKGA5K. Improving the scalability and the generalization of these approaches is left as future work, and we do not present their results here.

## 5.3 COMPARISON TO BASELINE ALGORITHMS

We use two metrics to compare algorithms. 1) *Average percent improvement over BRKGA 5K:* For a given graph, compute the percent improvement in the objective achieved by an algorithm relative to BRKGA with evaluation limit set to 5000. BRKGA 5K is a natural reference for measuring the effect of learning approaches that predict proposal distributions for it. 2) *Average percent gap from best known solution:* Compute the best known objective value among all the algorithms. (This will be found by CP-SAT if it finishes within the time limit.) Compute the percent difference between an algorithm's solution and the best known objective value. We report averages over test set graphs. Training set results are similar and reported in section A.3.

Table 1 compares REGAL to other algorithms on the two TensorFlow test sets and the synthetic dataset. REGAL outperforms all the baselines on all three tasks. It gives $1.9\times$ and $4.4\times$ bigger improvements

Table 1: Comparison of methods on the TensorFlow and Synthetic test sets. Results are averages over test set graphs. Higher is better for % Improvement over BRKGA5K, and lower is better for % Gap from best known. Note: CP-SAT, an enumerative algorithm, is run for up to 24 hours only to establish provably global optima (if possible) for evaluation purposes.

| | TF Runtime test set | | TF Peak Memory test set | | Synthetic Runtime test set | |
|---|---|---|---|---|---|---|
| Algorithm | % Improv. over BRKGA 5K | % Gap from best known | % Improv. over BRKGA 5K | % Gap from best known | % Improv. over BRKGA 5K | % Gap from best known |
| CP-SAT 24hr | 15.85% | 1.00% | -1.48% | 8.06% | 19.50% | 0.00% |
| GP + DFS | -37.32% | 66.98% | -6.51% | 14.77% | -55.8% | 93.66% |
| Local Search | -1.66% | 22.63% | 0.63% | 7.24% | 0.08% | 24.60% |
| BRKGA 5k | 0.00% | 20.19% | 0.00% | 7.98% | 0.00% | 24.63% |
| Tuned BRKGA | 3.20% | 16.40% | 0.80% | 7.11% | 3.11% | 20.76% |
| GAS | 3.79% | 15.24% | 0.16% | 7.67% | 0.80% | 23.48% |
| IDRS | -6.87% | 28.60% | -3.16% | 12.39% | -12.12% | 39.72% |
| **REGAL** | **7.09**% | **11.04**% | **3.56**% | **4.44**% | **4.81**% | **18.57**% |

than the next best algorithm on runtime and peak memory minimization tasks, respectively. The percent improvement over GP + DFS is 44.4% and 10.1% for runtime and peak memory, respectively. REGAL reduces the average percent gap from the best known solution by about $1.8\times$ with respect to BRKGA 5K on both TensorFlow test sets, and by about $6\times$ and $3.3\times$ with respect to GP + DFS on the TensorFlow Runtime and Peak Memory test sets, respectively. (For an XLA user, GP + DFS is the current, albeit weak, state-of-the-art algorithm.) The synthetic test set shows similar results. The learned policy successfully generalizes to previously unseen graphs, to the extent that a large fraction of the estimated room for improvement over BRKGA 5K is captured using the *same* evaluation limit.

To further test the limits of generalization for the policies learned using REGAL, we evaluate them on XLA graphs from a production compiler team's internal performance benchmark. XLA uses a different set of ops from TensorFlow, and the benchmark graphs on average have about an order of magnitude more nodes and edges than the TensorFlow graphs in our training set, so this is a difficult generalization challenge. REGAL achieves $0.58\%$ average runtime improvement over BRKGA 5K on 94 graphs, and $3.74\%$ average peak memory improvement on 32 graphs. It is promising that any improvements are possible at all despite training only on TensorFlow graphs, and points to the possibility of bigger improvements by training directly on XLA graphs.

**Optimizer running times:** BRKGA 5K takes on average 0.89 seconds on the TensorFlow Peak Memory test set to optimize a computation graph, while REGAL takes 1.04 seconds. (The times are similar on the Runtime test set.) Instead of taking hours to compute a solution per graph (e.g., Mirhoseini et al. (2017; 2018)), REGAL produces solutions in orders of magnitude less time, while still being better than all the baselines.

## 5.4 COMPARING REGAL VS. BRKGA

Figure 2 shows histograms of percent improvements in runtime (left) and peak memory (right) achieved by REGAL over BRKGA 5K on the test sets. Green bars correspond to graphs on which REGAL improved over BRKGA 5K, while red bars correspond to graphs on which REGAL was worse. (Ties have been omitted for clarity.) REGAL matches or beats BRKGA 5K on 87.4% of the runtime test set, and 88.9% of the peak memory test set. The highest improvement is 26.0% for run time and 54.3% for peak memory, while the worst regression is 24.0% for run time and 17.9% for peak memory.

To assess whether the improvements provided by REGAL's policy generalize to evaluation limits other than the one for which it was trained, we varied the evaluation limit used by both BRKGA and REGAL at test time. The results are shown in figure 3. REGAL's performance improves with more evaluations, confirming that the policy generalizes to higher evaluation limits. In other words, there exist node-level choices for the distributions used in BRKGA that perform well regardless of the evaluation limit, and REGAL learns to predict those choices. This is particularly useful in cases where the actual evaluation limit to use will be known only at test time, so that the same policy can be

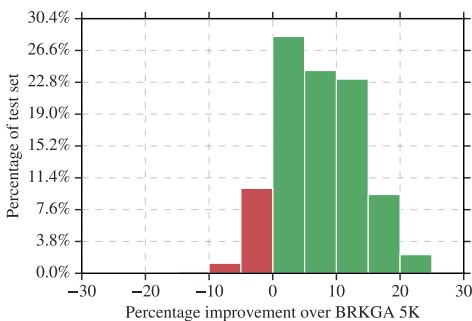 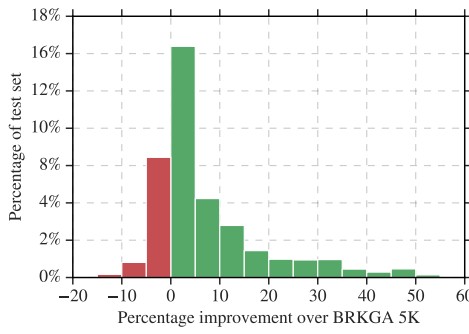

Fig. 2: Histogram of percent improvements in objective value on the TensorFlow runtime (left) and peak memory (right) datasets for test graphs on which REGAL is better (green) and worse (red) than BRKGA. (Ties are omitted from the figure for clarity but are included in the histogram percentage calculation.)

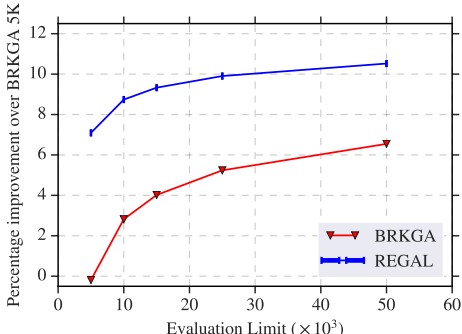 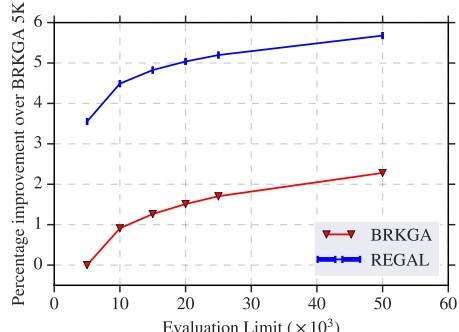

Fig. 3: Average percent improvement over BRKGA 5K given by REGAL and BRKGA on the TensorFlow test set for running time (left) and peak memory (right) as the evaluation limit is increased.

applied without re-training. Interestingly, even with 50K evaluations, BRKGA is not able to match REGAL's performance with just 5K evaluations!

## 5.5 GRAPH-DEPENDENT POLICY

The RL agent's actions are instance dependent. The agent that performs the best on the TF Runtime dataset has a choice of 16 different node placement actions for each node in a graph. For each graph in the TF Runtime test set, we compute the entropy of the distribution of the node placement actions taken by the agent and plot a histogram of these entropies in Figure 4(a). The mean of this distribution is 1.71 nats which implies that the actions are neither uniform random, nor constant, and vary from graph to graph.

Furthermore, the agent's performance overall gets better with more graph message passing iterations $T$. Figure 4(b) shows the peak validation reward reached within a hyperparameter sweep for each $T$ for the TF runtime optimization task. Models that utilize the GNN with message passing ($T > 0$) reach higher performance than $T = 0$ (i.e., ignoring the graph structure).

## 6 CONCLUSIONS AND FUTURE WORK

By training a graph neural network policy to predict graph-conditional node-level distributions for BRKGA, REGAL successfully generalizes to new graphs, significantly outperforms all baselines in solution quality, and computes solutions in about one second on average per TensorFlow test set

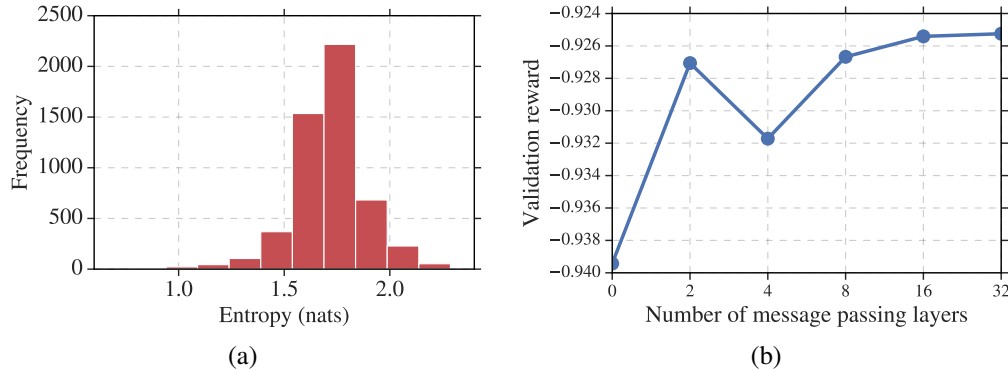

Fig. 4: The agent learns to utilize the graph structure. (a) The TF runtime agent picks a diverse set of actions. This plot shows the histogram of the entropy of the agent's actions across graphs in the dataset. (b) This plot shows the best validation reward achieved within a sweep of hyperparameters for each number of graph message passing rounds $T$. The performance gets overall better as $T$ increases, and models with $T > 0$ perform better than $T = 0$, which does not utilize the structure.

graph. REGAL's speed and generalization make it a strong choice for use in a production compiler that needs to handle a diverse set of graphs under a limited time budget.

We foresee several extensions. Integrating REGAL into a neural network compiler would allow us to evaluate the end-to-end gains due to better placement and scheduling decisions. To further improve REGAL's own performance, one could use a Mixture of Experts architecture. Given the diversity of graphs, a mixture model can train specialized sub-models on different types of graphs (e.g., convolutional networks, recurrent networks, etc.). Another is to replace BRKGA with alternatives, e.g., combining learned neural policies with local search.

ACKNOWLEDGMENTS

The authors would like to thank Ross Anderson, David Applegate, and Peter Hawkins for a significant amount of infrastructure on which this work builds, including the BRKGA and CP baselines, and Peter Ma, HyoukJoong Lee, and Peter Dolan for help with data collection.

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

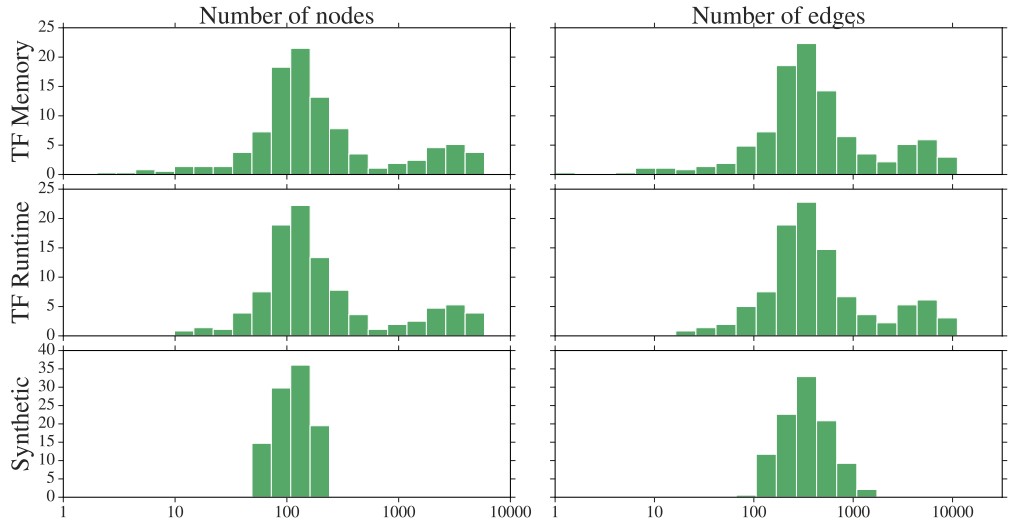

Fig. 5: Histograms of number of nodes (left) and edges (right) for the different datasets. The $y$-axis shows the percentage of graphs.

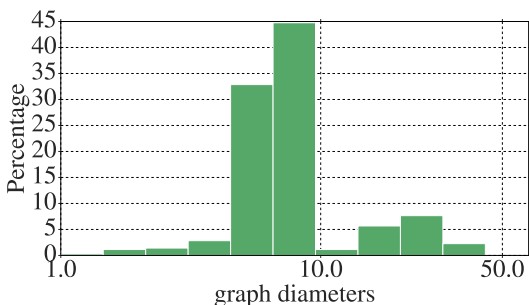

Fig. 6: Histogram of diameters of the graphs in the TF Runtime and TF Memory datasets.

## A APPENDIX

### A.1 DATASETS

#### A.1.1 DATASET STATISTICS

Figure 5 and figure 6 give statistics for the number of nodes and edges in the datasets. The broad range of graph sizes indicates the diversity of the datasets.

#### A.1.2 TENSORFLOW DATASET

We collected a dataset by mining TensorFlow jobs running in a shared production cluster and extracting computation graphs in the MetaGraphDef[2] format. As many computation graphs were repeated due to device/machine/job replicas, we de-duplicate the dataset by graph topology (specifically node in-degree sequence). We have not applied any other kind of filtering to restrict the dataset in any way (e.g., by architecture, input modality, learning task, dataset, etc.). Since the graphs were collected from a large and diverse set of production and research use cases across input modalities, learning types, and datasets, we strongly believe our dataset is representative of a broad, real-world distribution of TensorFlow graphs.

---

[2]`https://github.com/tensorflow/tensorflow/blob/master/tensorflow/core/protobuf/meta_graph.proto`

Computational costs for these computation graphs are simulated with an in-house simulator (based on Grappler[3]) that outputs memory and running time profiled information in the CostGraphDef[4] format. The simulator TensorFlow Op coverage didn't include custom kernels or complicated control flow Yu et al. (2018) like cycles (e.g. `tf.while_loop`).

The train-validation-test set split is made by selecting a set of 5 graphs from a list of graphs sorted by number of nodes, and splitting them as 3-1-1 across the three sets, respectively. This ensures that the distribution of the number of nodes is similar for the three sets.

For each graph in the train/validation/test sets, we make 99 copies of it and multiply each tensor size and each Op running time cost with a uniform sampled number in the interval $(0.5, 1.5)$ (one sample per tensor size per copy plus one sample per TF Op per copy). The modified copies are added back to the respective set so that the graph topologies in train/validation/test do not overlap.

Graphs with no relevant cost information or no room for improvement (e.g. a graph with a single node, a chain in the minimizing running time task) are filtered. This results in two different datasets for the two objective functions, one for runtime and another for peak memory.

The encoding is described in A.2

### A.1.3  XLA DATASET

We also collected a dataset by extracting CostGraphDefs during the XLA compilation of several benchmark graphs and extracted the greatest-size control-flow-free subgraph.

After deduplication 94 graphs remained.

The encoding is described in A.2

### A.1.4  SYNTHETIC DATASET

We sample synthetic graphs from a set of classic random graph models, including the Erdos-Renyi model Erdos & Rényi (1960), the Barabasi-Albert model Barabási & Albert (1999), the Watts-Strogatz model Watts & Strogatz (1998) and the stochastic block model Holland et al. (1983). The parameters we used for each of the random graph models are listed in Table 2.

| Model Type | Parameters |
|---|---|
| Erdos-Renyi | Edge probability $p = 0.05$. |
| Barabasi-Albert | Each node connected to $m = 2$ previous nodes. |
| Watts-Strogatz | Each node connected to $k = 4$ neighbors initially, with probability $p = 0.3$ to swap an edge with a random edge. |
| Stochastic Block Model | Number of blocks $k = 4$, within block edge probability $p = 0.3$, cross block edge probability $q = 0.01$. |

Table 2: Parameters for the random graph models used to generate synthetic graphs.

The number of nodes in a graph is sampled uniformly from range $[50, 200]$. Note that all the above random graph models generate undirected graphs. To convert the graphs into directed acyclic graphs, for each graph we sample a random ordering of nodes $\pi$, and then set the direction of all edges $(i, j)$ such that $\pi(i) < \pi(j)$, i.e. node $i$ comes before node $j$ in the ordering $\pi$. After setting all the edge directions, we locate all the head nodes that don't have any predecessors and all the tail nodes that don't have any successors, and then create a source node and a sink node, and create edges from the source node to all the head nodes, and from all the tail nodes to the sink node. Real TensorFlow graphs all contain one source and one sink node. Examples of synthetic graphs generated from the 4 random graph models are shown in Figure 7.

---

[3]https://github.com/tensorflow/tensorflow/tree/master/tensorflow/core/grappler

[4]https://github.com/tensorflow/tensorflow/blob/master/tensorflow/core/framework/cost_graph.proto

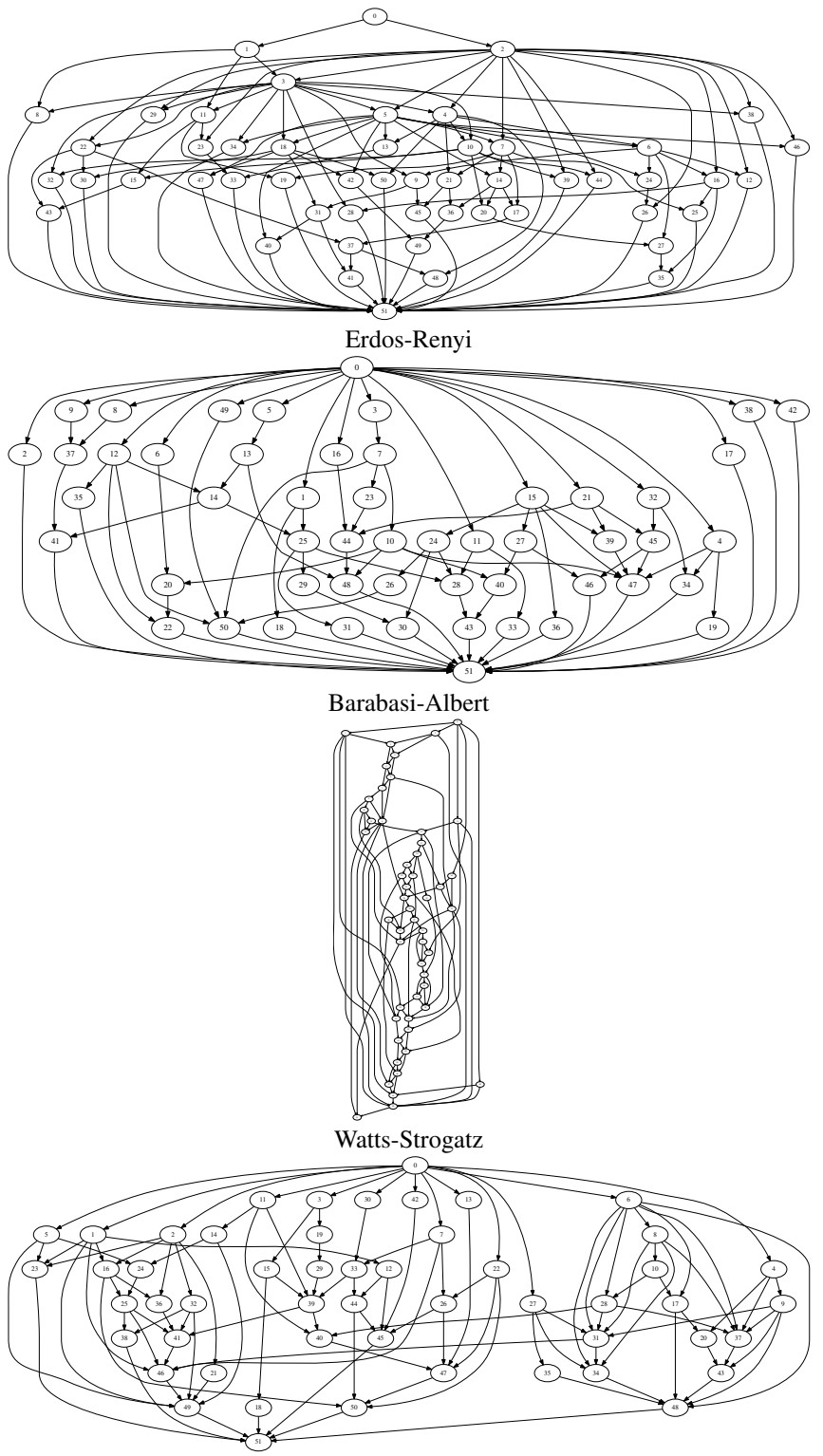

Fig. 7: Example synthetic graphs.

Each edge $(i, j)$ in the graph represents either a control dependency (op $j$ can run only when op $i$ is finished) or a data dependency (op $i$ can run only when some output tensor(s) produced by op $j$ is available). We assign a probability of 0.1, 0.8 and 0.1 for each op to produce 0, 1 or 2 output tensors. When op $i$ produces 0 output tensors, any other op $j$ that depends on it can only be through a control dependency. Otherwise, op $j$ will have a probability of 0.2 to make the dependency a control dependency, and probability of 0.8 to make this a data dependency, in which case an output tensor is picked according to an uniform distribution.

We fill in the memory cost for each tensor by sampling from a Gaussian distribution with mean 50 and standard deviation 10. The time cost for each op is computed as the sum of all the input and output tensor memory costs plus a random noise that is a fraction $r$ of the total memory cost, with $r$ sampled from a Gaussian with 0 mean and standard deviation 0.1. The source and sink nodes do not have any memory or time costs.

To make sure the generated synthetic graphs are interesting, we apply an additional filtering step by running BRKGA 1K and BRKGA 10K, and keeping only the graphs whose runtime improved by at least 18%. This gives us a dataset of graphs that on average improve runtime by 20% from running BRKGA 1K to 10K.

The synthetic data in CostGraphDef format is available at `https://github.com/deepmind/deepmind-research/tree/master/regal`.

The encoding is described in A.2.

## A.2   DATA ENCODING

We consider "control dependencies" as being tensors of size zero.

Each triple (op producer, tensor, op consumer) in the CostGraphDef is encoded as a separate edge. Each of these edges $e$ is associated with three features denoted by $x_e$: the size of the tensor that is associated with the edge, a one-hot feature indicating if the edge is a control edge, and normalized index of the hyperedge to which the edge belongs.

This means that the graph neural network's input is a directed graph with multiple edges. (There are alternative encodings like a bipartite graph where both TF ops and TF tensors are nodes, and edges exist only between ops and tensors when the op consumes or produces the tensor.)

Each node $v$ in the computation graph is associated with node features $x_v$. There are 11 node features in total, which can be grouped into three categories: memory-based, runtime-based (not used in the peak memory task) and BRKGA-based (task dependent).

As memory-based node features, we use the sum of input tensor sizes, the sum of output tensor sizes, the extra internal memory of the TensorFlow op, and a one-hot indicator of whether the op is the one which uses the greatest memory in the graph.

As runtime-based node features, we use the sum of direct predecessor nodes' running times, the sum of direct successor nodes' running times, the running time cost of the op itself, and one-hot indicator of whether the op is the one with the greatest runtime cost in the graph.

As BRKGA-based node features, we have a node aggregation (the expectation of the placement per device and the schedule order for each node) of the chromosomes found by BRKGA (minimizing peak memory for the peak memory dataset and minimizing runtime for the runtime dataset) running for 400 evaluations with uniform random distributions. To make comparisons fair, REGAL with $K$ fitness evaluations means $400$ evaluations to compute features, and $K - 400$ fitness evaluations for BRKGA using the instance-specific distributions.

For each graph, all node or edge features relating to memory size are normalized by the greatest memory size number in that graph and all features relating to runtime are are normalized by the greatest op runtime cost.

To break symmetry and reduce a single degree of freedom without loss of performance, we fix the placement of the node with highest memory for the memory task and runtime for the runtime task to the first device.

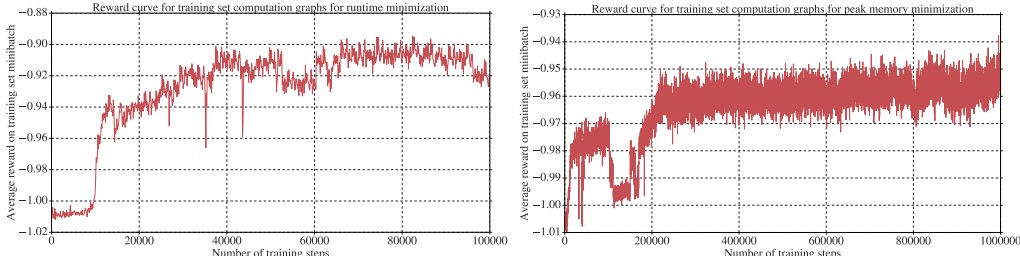

Fig. 8: Average reward achieved on a minibatch of training graphs during training for runtime minimization (left) and peak memory minimization (right)

## A.3 TRAINING SET RESULTS

Figure 8 shows the reward curves on the training set for runtime minimization (left) and peak memory minimization (right). Each point in the curves is the average reward achieved on a minibatch of training graphs at the corresponding training step. The graph neural network policy is trained using TensorFlow on 10 cores of a CPU machine (no GPUs were used), with multi-threading used by BRKGA for evaluating chromosomes and by TensorFlow. A training run takes approximately 2-3 days to be completed. The final average percent improvement over BRKGA5K on the training set is similar to that of the test set for both tasks: $7.25\%$ on the training set vs. $7.09\%$ on the test set for runtime minimization, and $4.36\%$ on the training set vs. $3.56\%$ on the test set for peak memory minimization. The small gap between train and test results shows that the policy is able to generalize successfully to unseen graphs at test time.

## A.4 PERFORMANCE MODEL

The scheduling problem is specified by a set of devices $D$ and a computation graph. The computation graph has the list of *ops* $j \in N$ and *tensors* $\tau \in T$, the tensors produced by each op $I(j) \subseteq T$, the tensors consumed by each op $C(j) \subseteq T$, the memory used by each tensor $m_\tau$, and the execution time of each op $r_j$. A tensor is produced by exactly one op but can be consumed by many.

Solutions to the scheduling problem are constructed as follows. A placement is an assignment $pl : N \to D$ from ops to devices. Given a placement we define $\tilde{N}_{pl}$ as the set of ops $N$ extended with synchronous inter-device transfer operations. A transfer operation consumes a tensor on the device where it was created and produces it on a device where it is needed.

Given a placement $pl$, a schedule is a total ordering $s : \tilde{N} \to \{1, 2, \dots, |\tilde{N}|\}$ on ops in $\tilde{N}_{pl}$. We say that op $j$ runs at simulation time step $s(j)$. We model the schedule execution as follows. At each simulation time step $t \in \{1, 2, \dots, |\tilde{N}|\}$, each device $d$ has a list $l_{d,t}$ of tensors currently in memory. A tensor is added to the list when produced by an op that runs on the device or by a transfer op that receives the tensor on the device. A tensor is removed immediately after all of its consumers on the device have run. A schedule is valid if for each op $j$, all the input tensors are available on the corresponding device at simulation time step $s(j)$. See Section A.9 for an example schedule.

The memory used on a device at simulation time step $t$ is the sum of the memory used by each tensor that is in memory, i.e., $\sum_{\tau \in l_{d,t}} m_\tau$. The peak memory of a schedule is the maximum value of the memory used at any time and on any device.

The runtime of a schedule is computed by stepping through the simulation time steps in order and accounting for the execution time of each op on each device. Synchronous transfers block until both the sender and the receiver have completed their preceding tasks, and their execution time depends on a known bandwidth between devices.

## A.5 BRKGA CHROMOSOME ENCODING

Let the input graph $G$ contain $o$ ops and $t$ tensors which must be placed over $d$ devices. Then, the BRKGA chromosome for this graph is a vector $c \in [0, 1]^{o \times d + o + t \times d}$ composed of the following parts

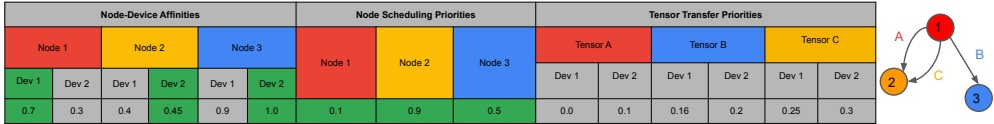

Fig. 9: A computation graph and an example of a chromosome encoding.

1. The first $o \times d$ entries in $\boldsymbol{c}$ represent the node-device affinities, one value for each (node, device). Each node is assigned to the device for which it has the highest value in the chromosome.

2. The next $o$ entries represent the node scheduling priorities. A valid schedule of the computation graph is obtained by performing a topological sort over the nodes of the graph, breaking ties using the node scheduling priorities. Nodes with higher priority are scheduled first.

3. The final $t \times d$ entries represent the tensor transfer priorities, one entry for each (tensor, device) pair. These priorities determine the order in which tensors are transferred across devices.

An example of a chromosome encoding is shown in Figure 9 for a graph with $o = 3$ nodes, $t = 3$ tensors and $d = 2$ devices. As per the example, nodes 1 and 3 are placed on device 1 while node 2 is placed on device 2. The scheduling order over the nodes is 1, 2, 3. Since nodes 1 and 2 are placed on different devices, tensors A and C must be transferred from device 1, where they are produced, to their consumer, node 2, which is on device 2. As per the tensor transfer priorities, tensor C is transferred before tensor A since tensor C has a higher priority to get transferred to device 2.

## A.6 ACTION DEQUANTIZATION

Each real number in a BRKGA chromosome is sampled from its own Beta distribution, which is parameterized by two real numbers $\alpha$ and $\beta$. To be more precise, if we denote the chromosome by $\boldsymbol{c} \in \mathbb{R}^L$, then $c_i \sim \mathcal{D}(\alpha_i, \beta_i) \,\forall\, 1 \leq i \leq L$ where $\mathcal{D}(\alpha_i, \beta_i)$ is a Beta distribution with parameters $\alpha_i$ and $\beta_i$. To be able to run BRKGA, REGAL must propose the values for $\alpha_i$ and $\beta_i$ for each $i$.

As described in A.5, the BRKGA chromosome consists of three parts. In REGAL, we optimize the RL agents over choices of $\alpha_i$ and $\beta_i$ for the first two parts of the chromosome, i.e., the parts of the chromosome corresponding to the node placement and scheduling decisions. The Beta distribution parameters for the tensor transfer priorities are fixed to $\alpha_i = \beta_i = 1$ which correspond to the uniform random distribution. Thus, for a graph with $o$ ops and with $d$ devices, the RL agent must propose $(d + 1) \times 2$ values for each of the $o$ ops in the graph.

To make the learning task easier, rather than directly predicting values of $\alpha$ and $\beta$, we quantize the output space of the RL agent's actions such that each action uniquely maps to a Beta distribution. This mapping is done as follows:

- For each node in the graph $1 \leq i \leq o$, and for each entry $1 \leq j \leq (d + 1)$ corresponding to node $i$ in the BRKGA chromosome, the agent performs the set of actions

$$\boldsymbol{a}_i = \{m_{i,1}, v_{i,1}, m_{i,2}, v_{i,2}, \ldots, m_{i,j}, v_{i,j}, \ldots, m_{i,d+1}, v_{i,d+1}\}.$$

- $m_{ij}$ and $v_{ij} \in \{0, 1, \ldots, k - 1\}$ where $k$ is some fixed constant greater than 1, and these represent the quantized mean $\mu_{ij}$ and variance $\sigma_{ij}^2$ of the Beta distribution which are related to each other as follows:

$$\mu_{ij} = \frac{m_{ij} + 1}{k_{ij} + 1}, \quad \sigma_{ij}^2 = \mu_{ij} \times (1 - \mu_{ij}) \times \frac{v_{ij} + 1}{k_{ij} + 1}$$

- $\mu_{ij}$ and $\sigma_{ij}^2$ can be mapped to $\alpha_{ij}$ and $\beta_{ij}$ for a Beta distribution as follows:

$$\beta_{ij} = \mu_{ij} \times \frac{(1 - \mu_{ij})^2}{\sigma_{ij}^2} - 1 + \mu_{ij}, \quad \alpha_{ij} = \beta_{ij} * \frac{\mu_{ij}}{1 - \mu_{ij}}$$

- The values $m_{ij}$ and $v_{ij}$ are sampled from a Categorical distribution whose logits are determined by $\mathrm{MLP}_a(\boldsymbol{h}_i)[k * (j-1) : k * j]$.

We use a similar quantization strategy for the BRKGA crossover probabilities. For every crossover probability, we sample an integer $c \in \{0, 1, \ldots, k-1\}$ from a Categorical distribution for some fixed integer constant $k$, and the dequantized crossover probability is given by $0.5 * \left(1 + \frac{c+1}{k}\right)$

## A.7 EXTRA MODEL DETAILS

**MLPs** Multi-layer perceptrons, or multi-layer fully connected neural networks are models that map input vectors to output vectors through layers of linear transformations and nonlinear activation functions, like the following:

$$\boldsymbol{h} = \mathrm{MLP}(\boldsymbol{x}) = \boldsymbol{W}_l \sigma_{l-1}(...\sigma_2(\boldsymbol{W}_2 \sigma_1(\boldsymbol{W}_1 \boldsymbol{x} + \boldsymbol{b}_1) + \boldsymbol{b}_2))...) + \boldsymbol{b}_l, \tag{3}$$

where $\boldsymbol{x}$ is an input vector, $(\boldsymbol{W}_i, \boldsymbol{b}_i)$ are the parameters for the $i$th layer, and $\boldsymbol{h}$ is the output vector. $\sigma$ is a nonlinear scalar function applied element-wise to the input vectors. Typical choices include the logistic sigmoid function $\sigma(x) = \frac{1}{1+e^{-x}}$, tanh function $\sigma(x) = \frac{e^x + e^{-x}}{e^x - e^{-x}}$ and the ReLU function $\sigma(x) = \max\{0, x\}$.

**RNNs and LSTMs** Recurrent neural networks (RNNs) are good sequence models. Typical RNNs contains a recurrent memory $\boldsymbol{c}_t$ that is updated recursively by taking some input at each step $t$ as the following:

$$\boldsymbol{c}_t = \mathrm{RNNCell}(\boldsymbol{c}_{t-1}, \boldsymbol{x}_t), \tag{4}$$

where $\boldsymbol{x}_t$ is the input at step $t$. The simplest RNN cell has the following form

$$\boldsymbol{c}_t = \sigma(\boldsymbol{W}[\boldsymbol{c}_{t-1}, \boldsymbol{x}_t] + \boldsymbol{b}), \tag{5}$$

where $\boldsymbol{W}, \boldsymbol{b}$ are the parameters and $\sigma$ is a nonlinearity.

Long-short term memory (LSTM) models are a type of RNNs that uses explicit gating to control the access to the memory. LSTMs distinguish the memory $\boldsymbol{c}_t$ and the output of the LSTM $\boldsymbol{h}_t$ as two sets of vectors, and compute the update at step $t$ as

$$\boldsymbol{i} = \mathrm{sigmoid}(\boldsymbol{W}_i[\boldsymbol{h}_{t-1}, \boldsymbol{x}_t] + \boldsymbol{b}_i) \tag{6}$$

$$\boldsymbol{f} = \mathrm{sigmoid}(\boldsymbol{W}_f[\boldsymbol{h}_{t-1}, \boldsymbol{x}_t] + \boldsymbol{b}_f) \tag{7}$$

$$\boldsymbol{o} = \mathrm{sigmoid}(\boldsymbol{W}_o[\boldsymbol{h}_{t-1}, \boldsymbol{x}_t] + \boldsymbol{b}_o) \tag{8}$$

$$\boldsymbol{g} = \tanh(\boldsymbol{W}_g[\boldsymbol{h}_{t-1}, \boldsymbol{x}_t] + \boldsymbol{b}_g) \tag{9}$$

$$\boldsymbol{c}_t = \boldsymbol{f} \odot \boldsymbol{c}_{t-1} + \boldsymbol{i} \odot \boldsymbol{g} \tag{10}$$

$$\boldsymbol{h}_t = \boldsymbol{o} \odot \tanh(\boldsymbol{c}_t). \tag{11}$$

Here $\boldsymbol{i}, \boldsymbol{f}, \boldsymbol{o}$ are the input, forget and output gates and $\odot$ is element-wise multiplication. The carefully designed memory access control through gating makes LSTMs better at modeling long-term dependencies.

## A.8 AUTOREGRESSIVE PREDICTION MODELS

We can use an autoregressive model to capture structure in the outputs. Given the node representations $\boldsymbol{h}_v$ for each of the nodes from the GNN, we can utilize an ordering of the nodes, e.g. from a topological sort, and treat the node representations as a sequence, and then use an LSTM (Hochreiter & Schmidhuber, 1997) to predict the outputs $\boldsymbol{y}_v$ sequentially.

We tried this approach but found that using an LSTM on top of the $\boldsymbol{h}_v$'s to predict $\boldsymbol{y}_v$'s did not perform as well as the conditionally independent model. The reasons for this might be: (1) the autoregressive approach relies on a sequential ordering of the nodes, and this ordering might not be reliable nor consistent across graphs; (2) the number of nodes in the computation graphs can be large, and learning recurrent models on long sequences is known to be challenging (Pascanu et al., 2013); (3) the noisy training signal in our REINFORCE-based training setup makes this model even more difficult to train.

### A.9 PLACEMENT AND SCHEDULING EXAMPLE

Figure 10 illustrates a computation graph, a valid schedule, and how we account for which tensors are in memory at a given time under the model presented in sec. 3.1.

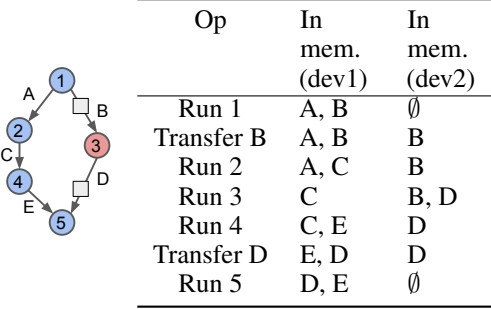

| Op | In mem. (dev1) | In mem. (dev2) |
|---|---|---|
| Run 1 | A, B | $\emptyset$ |
| Transfer B | A, B | B |
| Run 2 | A, C | B |
| Run 3 | C | B, D |
| Run 4 | C, E | D |
| Transfer D | E, D | D |
| Run 5 | D, E | $\emptyset$ |

Fig. 10: An example computation graph and execution schedule across two devices. Op 3 is assigned to device 2 while all others are assigned to device 1.

### A.10 BASELINES

Here we provide supplementary information about our baselines.

- CP SAT: The multi-device peak memory minimization problem is formulated for the CP solver using a model of the operation execution order, tensor lifetimes, and cumulative constraints to evaluate peak memory usage. The solver is guaranteed to find the globally optimal solution given sufficient time.

- Graph partition + DFS: The graph partitioning objective is to minimize data transferred across devices. We use the modified implementation of the Kernighan-Lin algorithm (Kernighan & Lin, 1970) used by XLA for device placement in some settings. This implementation is generally slower than heuristics implemented in popular libraries like METIS (Karypis & Kumar, 1998) although it tends to find better quality solutions.

- Local Search: The initial schedule is a topological sort order of the ops, and the initial placement selects devices uniformly randomly. A local move either changes the device assignment of the op, or changes the op order in the current schedule. The hyperparameters (e.g., number of random restarts) are set to values that perform the best on a sample of 10,000 graphs in the training set as found by grid search.

- Tuned BRKGA: The following hyperparameters of BRKGA using grid search: the beta distribution parameters (two scalars), and the number of chromosomes, elites, mutants, and populations. The grid search tries 648 hyperparameter settings and picks the best one as evaluated on 10,000 training set graphs.

- REGAL: The performance of REGAL is stochastic both because the actions are stochastic and because BRKGA itself depends on random samples for mutation and crossover. We estimated the standard deviation of the percent improvement statistics with respect to these sources of randomness as below 0.1%, which is small compared to the differences we observe. Hence we have omitted the error bars from figures 3.

### A.11 RUNNING TIME COMPARISON FOR ALGORITHMS

Table 3 shows the average running times of the various algorithms on the TensorFlow test set and the XLA dataset, as measured on an Intel Xeon E5-1650 3.60GHz machine. The times are averaged over the unaugmented graphs in the test set. REGAL provides both fast running time as well as high solution quality. For a slightly higher running time than BRKGA 5K, REGAL improves the solution quality significantly. Almost all of the added running time is due to extra cost of sampling beta distributions by REGAL compared to uniform distributions by BRKGA. This can be seen from the nearly identical running times of REGAL and Tuned BRKGA, which also uses beta distributions,

Table 3: Average running times for all methods (time cost of running the algorithms, not to be confused with solution quality).

| Algorithm | TF Peak Memory test |
|---|---|
| CP SAT | ~2 hours |
| GP + DFS | 144 sec |
| Local Search | 122 sec |
| BRKGA 5K | 0.89 sec |
| Tuned BRKGA | 1.04 sec |
| GAS | 1.04 sec |
| REGAL | 1.04 sec |

Table 4: Performance of REGAL on peak memory with various subsets of actions.

| Placement | Scheduling | Valid | Test |
|---|---|---|---|
| Yes | No | -0.4% | -0.2% |
| No | Yes | 4.4% | **3.65%** |
| Yes | Yes | **4.67%** | 3.56% |

but without the neural network policy. The local search heuristic runs slowly because it was not implemented efficiently, e.g., with incremental evaluations of the objective; we show its timing results for completeness only.

## A.12 ABLATION ANALYSIS OF AGENT ACTION TYPES

REGAL can train a policy to generate any subset of the following actions for BRKGA: 1) Actions for node placement priorities, and 2) Actions for node scheduling priorities. We train REGAL with various subsets of these actions and compare their performance against each other in table 4.

We observe that on the validation set, REGAL performs best when it has to learn actions for both placement and scheduling compared to just scheduling or placement alone.

## A.13 ANALYSIS OF THE STRUCTURE OF PLACEMENT AND SCHEDULING DECISIONS,

Our best model trained on the TF peak memory dataset is capable of generating 16 different kinds of actions for node placement decisions and 4 different kind of actions for node scheduling decisions. Each of these actions determine the shape of the Beta distributions from which we sample the node-device affinities and the node scheduling priorities. In this section we attempt to gain insights into the structure of these actions.

We divide our node placement decisions into three categories:

- Actions that give a node a higher probability to be placed on device 1
- Actions that give a node a higher probability to be placed on device 2
- Actions that give equal preference to the two devices.

Similarly, we divide our node scheduling decisions in two categories:

- Actions that give nodes a "high" scheduling priority.
- Actions that give node a "low" scheduling priority.

Finally, we aggregate the average relative memory consumption of all nodes that were assigned the same set of actions, where memory consumption of a node is defined as the sum of the memory uses of all its input and output tensors. The relative memory usage is the memory usage normalized by the largest memory usage of a node in the graph.

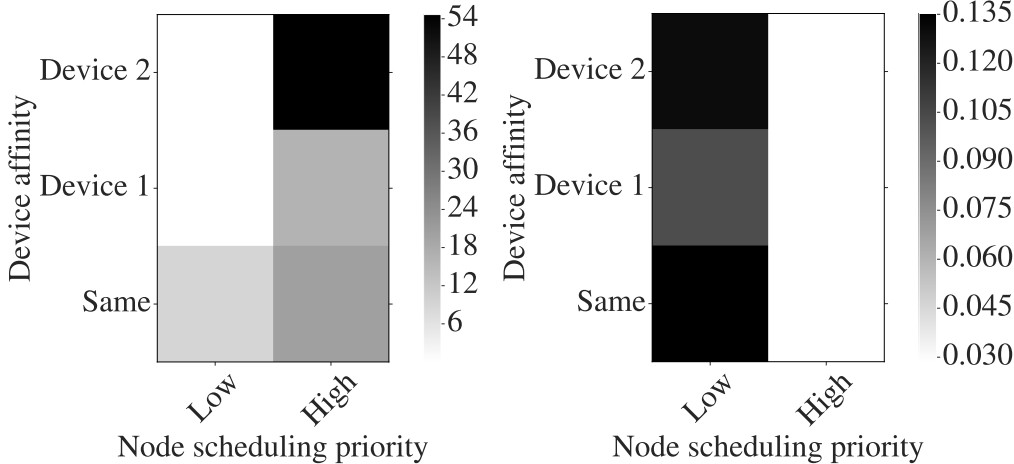

Fig. 11: The agent picks a diverse set of actions. These plots show the frequency of actions chosen (in percent, left) and the average relative weights of the nodes that the actions are applied to (right)

We plot this data in Figure 11. On the right, each cell represents the average relative memory consumption of the nodes that were assigned a particular placement and scheduling decision (darker cells indicate nodes with higher average relative memory consumption). On the left, each cell represents the frequency of the nodes that were assigned a particular placement and scheduling decision (darker cells indicate higher frequency). With these we can make the following observations:

- On average, nodes with higher normalized memory consumption are assigned lower scheduling priorities.

- Most of the nodes with the highest relative memory consumption have no affinity for either of the two devices.

- For nodes with the lowest relative memory consumption, most of them have an affinity to be placed on device 2, while a smaller but still significant number of them prefer device 1.

This implies that the node placement strategy is more complicated than a trivial "place lighter nodes on device 2" strategy and REGAL's actions are non-trivially dependent on the input.

### A.14 HYPERPARAMETERS OF THE BEST AGENT FOR PEAK MEMORY TF

The graph neural network had a state size of 32 for each node and edge, 16 propagations, all networks $\text{MLP}_n$ $\text{MLP}_e$ $\text{MLP}_{\text{node}}$ $\text{MLP}_{\text{msg}}$ $\text{MLP}'_{\text{msg}}$ being two layers of size 32, the aggregation used was mean pooling. For faster training, the reward of the training set was made with 1000 fitness evaluations for REGAL and BRKGA (4600 for REGAL and 5000 for BRKGA for the validation and test sets). Training lasted 100000 gradient steps with each step having a mini-batch of size 4 and with gradient clipping by $L_2$ norm with value 10 . The baseline mean squared error term's contribution to the overall loss was weighted by 0.0001 . The optimizer was Adam with beta1 0.9 beta2 0.999 epsilon $1e-8$ and learning rate 0.0001 . The number of devices (for the memory model) was 2.

### A.15 HYPERPARAMETERS OF THE BEST AGENT FOR RUNTIME TF

The graph neural network had a state size of 32 for each node and edge, 16 residual graph propagations, all networks $\text{MLP}_n$ $\text{MLP}_e$ $\text{MLP}_{\text{node}}$ $\text{MLP}_{\text{msg}}$ $\text{MLP}'_{\text{msg}}$ being two layers of size 32, the aggregation used was sum. Training lasted 100000 gradient steps with each step having a mini-batch of size 4 and with gradient clipping by $L_2$ norm with value 10. The baseline mean squared error term's contribution to the overall loss was weighted by 0.0001 . The optimizer was Adam with beta1 0.9 beta2 0.999 epsilon $1e-8$ and learning rate 0.0001 . With $k=16$ for scheduling and $k=2$ for placement (k being the quantization level defined in A.6).

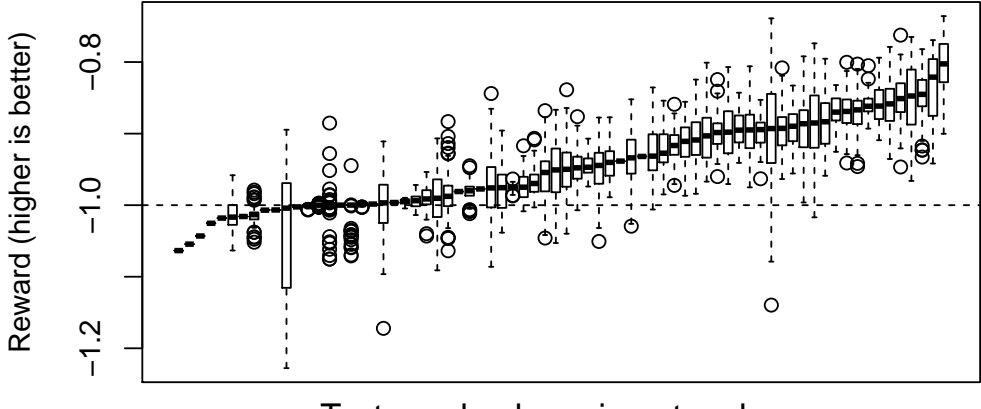

Fig. 12: A box plot of rewards on the TF Runtime test set by unique graph topology. There are 100 graphs for each topology, 99 of which generated by data augmentation. A reward greater than -1 implies that REGAL finds a better solution than BRKGA. Box plots visualize the 25th, 50th, and 75th percentiles and the range of a set of points.

### A.16 HYPERPARAMETERS OF THE BEST AGENT FOR RUNTIME SYNTHETIC

Same as A.15 but with 2 graph propagations and GRU node updates and aggregation using mean.

### A.17 PERFORMANCE BY GRAPH TOPOLOGY

Figure 12 shows the distribution of rewards on the TF Runtime test set broken down by graph topology. As described in Section 5.1, for each unique topology we augment the dataset by perturbing the tensor sizes and op running times. This generates a distribution of 100 rewards per topology. The variance for a fixed topology is typically relatively small.

