# OpenReview forum: "Reinforced Genetic Algorithm Learning for Optimizing Computation Graphs"
_ICLR.cc/2020/Conference — Accept (Poster)_

### Official Review · AnonReviewer1 · 2019-10-24
**Official Blind Review #1**

**Rating:** 6

**Review:**

In this work the authors propose a deep RL approach to minimize the makespan and the peak memory usage of a computation graph as produced by MXNet/PyTorch/TensorFlow.  This is an increasingly important problem as distributed deep learning is necessary in many cases. The authors aim to minimize the execution time and/or the peak memory usage. For this purpose they generate a training dataset out of a real-world dataset of various TensorFlow computation graphs using simulation software. The proposed RL approach consists of two steps. First  a GNN is used to derive representations for computation graphs. Then the authors use a heuristic BRKGA to learn a policy for the placement of computation graphs, that actually works on unseen graphs. Overall this paper is well-written, deals with an important practical problem. While it is not immediately clear to the reader the effect of BRKGA on the mapping of the graph to the resource network and why it works so well, the results are convincing (but still there is space for improvement).  That is why I rate it as a "weak accept".

- Can you explain why the beta distribution choices at each node may have a negative impact on the makespan in certain cases? Have you looked into them?
- To what extent are the simulations realistic? Can you please comment more on this aspect?
- Have you tried Scotch? https://www.labri.fr/perso/pelegrin/scotch/. Since the software aims to achieve a different objective, it serves as a baseline.
- Can you obtain insights with respect how you could cluster the TensorFlow computation graphs?
- Can you improve the discussion on BRKGA? Since it is a vital component of the proposed framework, it would be informative to read few more self-contained details on how it works in section 2.
- Once you obtain a mapping, can you comment on any insights concerning the structure of the partition and schedule?

**Experience Assessment:**

I have published one or two papers in this area.

**Review Assessment: Checking Correctness Of Derivations And Theory:**

I assessed the sensibility of the derivations and theory.

**Review Assessment: Checking Correctness Of Experiments:**

I carefully checked the experiments.

**Review Assessment: Thoroughness In Paper Reading:**

I read the paper thoroughly.

---

> ### Author Response · Authors · 2019-11-14
> **Response to "Official Blind Review #1"**
>
> Thank you for the review and the interesting questions!
>
> > Then the authors use a heuristic BRKGA to learn a policy ..., that actually works on unseen graphs.
> > ...it is not immediately clear to the reader the effect of BRKGA on the mapping of the graph to the resource network and why it works so well, …
>
> To clarify, BRKGA is a genetic algorithm we use to solve the joint placement and scheduling problem. BRKGA guides its search based on the solutions seen so far, like a classical optimization algorithm. We introduce learning by training a Graph Neural Network (GNN) that defines a mapping from computation graphs to mutant sampling distributions for BRKGA. The combination of GNNs and BRKGA is REGAL.
>
> We expanded the description of BRKGA in Section 3.2 to help clarify its role. It is challenging to explain why it works so well. In addition to the references cited in the paper, we recommend the tutorial given by Resende at CLAIO/SBPO 2012 (http://mauricio.resende.info/talks/2012-09-CLAIO2012-brkga-tutorial-both-days.pdf ). BRKGA is a relative of the cross-entropy method (https://doi.org/10.1007/s10479-005-5724-z ) that has been successfully applied in combinatorial optimization and machine learning.
>
> Note also that to apply BRKGA to a specific problem, one must design a mapping from [0, 1]^n to the space of solutions. An exploration of design choices here could yield insights but is outside the scope of this work.
>
> > - Can you explain why the beta distribution choices ... have a negative impact on the makespan in certain cases? ...
>
> We don’t have detailed insights for why the learned policy performs worse than BRKGA for certain cases. However, it is easy to formulate an example where this can occur—consider a mutant sampling distribution that has unit probability mass at a single poor solution. In such a case, REGAL (i.e., BRKGA with this bad sampling distribution) will never sample good solutions, but plain BRKGA may find better solutions by using the uniform random distribution.
>
> > - To what extent are the simulations realistic? ...
>
> We have validated our performance model in an end-to-end production setting that is more restricted than the setting in the paper. When the number of devices (i.e., d) equals 1, the performance model reliably identifies schedules with low peak memory usage. The runtime part of the simulation, only non-trivial when d > 1, has not yet been validated with experiments on hardware. We expect that it will be necessary to model the asynchronous aspect of transfers in order to accurately predict runtimes on real hardware.
>
> Rather than claiming that the performance model is realistic, we have claimed that it provides a challenging (i.e., NP-hard) setting in which to study how to learn an optimizer. Maintaining the simpler performance model also allows us to compare with baselines like constraint programming (CP), which help us validate the methodology. While CP would be hard to extend to more complex performance models, REGAL can be applied just as well.
>
> > - Have you tried Scotch? …
>
> We have not tried Scotch; however, our GP+DFS baseline is analogous to Scotch, to the best of our knowledge. We set up the graph partitioning (GP) problem as follows: Each node in the graph is a TensorFlow operation, and edges represent direct data dependencies, with weights proportional to the sizes of the tensors. We aim to find a partitioning of the nodes into d (= 2) disjoint subsets such that the weight of edges that cross the subsets is minimized. We believe that this matches the graph partitioning setup, e.g., reported in Mirhoseini et al. (ICML ‘17).
>
> In place of Scotch, we use an implementation of the classical Kernighan–Lin algorithm modified to support weighted edges. We chose this proprietary implementation over Scotch because it’s already in use in an optimizing compiler for device-placement decisions.
>
> > - Can you obtain insights with respect how you could cluster the TensorFlow computation graphs?
>
> We have not yet tried to obtain insights about how the policy’s behavior can be used to cluster computation graphs. One possibility is to learn a fixed dimensional graph-level embedding as part of the policy network, and then to cluster the embedding vectors for the training set graphs. Another possibility, as mentioned in Section 6, is to use a Mixture of Experts architecture for the policy, and once trained, analyze which graphs are selected by which experts to understand how the policy clusters graphs. Both of these are interesting directions for future work.
>
> > - Can you improve the discussion on BRKGA? …
>
> See changes to Section 3.2.
>
> > - ... can you comment on any insights concerning the structure of the partition and schedule?
>
> We have added a new section (A.13), where we provide some insights into the structure of the joint placement and scheduling policy at the node-level. While we see some patterns in Figure 10, the overall learned policy remains non-trivial.

---

### Official Review · AnonReviewer3 · 2019-10-28
**Official Blind Review #3**

**Rating:** 6

**Review:**

Summary

This paper proposes an ML-based method to optimize TensorFlow Graph execution. Specifically, it combines graph neural networks (GNNs) and BRKGA (a genetic algorithm) to search over the joint space of TF node-device placement and scheduling. The core claims on the advantages of this method are that (1) it co-searches placement and scheduling space, (2) the trained model can generalize to different graphs and inference cost is very  small. The experimental results show that REGEL can outperform a few baseline methods on this problem.

Writing
- The paper is well-written and I enjoyed reading the paper.
- Some more descriptions about the BRKGA algorithm could be added in.


Method and Results

Some confusion if the authors could answer:
- I am very confused by one of the claims that “ the first work on learning a policy for jointly optimizing placement and scheduling”. I don’t see much evidence in the result section about showing the co-searching the joint space yield advantages? I am fairly familiar with the line of work on only optimizing device placement, but it would be good to see some ablation studies showing search over the joint space is advantageous.

- The model is trained with standard REINFORCE -- how many training time and resources are needed to train a REGEL model for a task? How’re the training dynamics looking like (variance, convergence, etc?)?

- In terms of the generalization ability of REGEL, the paper has clearly shown that REGEL is able to generalize to differently shaped graphs, with acceptable cost, but I am wondering for the same dataflow graph, how REGEL generalizes to different input data configurations (size, modality, etc.)? E.g. if the batch size of the input data is changed, the execution time of each kernel and their memory usage (in general, the system treatment) would change; Can a trained REGEL model on a data config A generalize to B? How would this affect the performance of REGEL?

- It seems the method and assumptions about graphs or training data are pretty coupled with TensorFlow and graph-mode execution, how could the method be generalized to other ML frameworks (e.g. frameworks with eager execution)

- Could the authors clarify why the two methods mentioned in “Learning to directly predict a solution” has quadratic complexity w.r.t. # of nodes and whereas REGEL is linear?

- Confusion on Figure 4(b): Could some more critical statistics about the graphs in the training/test dataset be reported? e.g. what’s the average depth of the training graphs? When there are 32 MP layers a node’s feature will be passed across its 32-hop neighborhood, which seems surprising as it is common to observe GNN starts degenerating with increased depth (because all node features become similar during message passing)


**Experience Assessment:**

I have published one or two papers in this area.

**Review Assessment: Checking Correctness Of Derivations And Theory:**

I carefully checked the derivations and theory.

**Review Assessment: Checking Correctness Of Experiments:**

I carefully checked the experiments.

**Review Assessment: Thoroughness In Paper Reading:**

I read the paper thoroughly.

---

> ### Author Response · Authors · 2019-11-15
> **Response to "Official Blind Review #3"**
>
> > The paper is well-written and I enjoyed reading the paper.
>
> Thanks for your comments!
>
> > Some more descriptions about the BRKGA algorithm ...
>
> See changes to Section 3.2.
>
> > - I am very confused by one of the claims that “ the first work on learning a policy for jointly optimizing placement and scheduling”. …
>
> Good point. Our claim was not clearly stated, and we have changed the discussion in the introduction. The works we cite that learn a policy for device placement relied on TensorFlow’s dynamic scheduler to make the scheduling decisions. In that setting, we do not claim that one should jointly optimize placement and scheduling, and it’s not obvious how to do so.
>
> On the other hand, we have approached the problem from the perspective of static scheduling, which applies in a number of recently developed compilers for deep learning computation graphs. In the static setting, jointly deciding the assignment from the operations to devices and the schedule of operations within a device is a classical problem and hence a natural one to solve; see Kwok and Ahmad (1999) and Sinnen (2007), both cited in our paper. Deciding on placement and scheduling separately makes the task harder. The poor performance of the GP+DFS baseline is an example of this; the graph partitioner ignores scheduling when making placement decisions. A similar motivation for joint optimization can also be found for the problem of CPU instruction scheduling and register allocation, see e.g., Motwani et al. (1995) https://pdfs.semanticscholar.org/1b7d/20b856fd420f93525e70a876853f08560e38.pdf.
>
> We have indeed performed ablation tests on the value of learning in the placement and scheduling spaces; see appendix section A.12. These results suggest that the majority of gains from REGAL are in fact thanks to learning better sampling distributions in the scheduling part of the action space.
>
> > - The model is trained with standard REINFORCE -- how many training time and resources are needed to train a REGEL model for a task? How’re the training dynamics looking like (variance, convergence, etc?)?
>
> See figure 7 in section A.3 for the training set reward curves for runtime and peak memory minimization tasks. The graph neural network policy is trained using 10 cores of a CPU machine (no GPUs were used), with multi-threading used by BRKGA for evaluating chromosomes and by TensorFlow. A training run takes approximately 2-3 days to be completed.
>
> > - In terms of the generalization ability of REGEL, the paper has clearly shown that REGEL is able to generalize to differently shaped graphs, with acceptable cost, but I am wondering for the same dataflow graph, how REGEL generalizes to different input data configurations (size, modality, etc.)? …
>
> We have not tested REGAL in this setting, opting to focus on the harder task of generalizing across graphs with different topologies. It would indeed be interesting to see if the performance gains are greater when the dataset is restricted to variations on a single topology.
>
> To partially address this question, we performed an additional analysis on our results. We’ve added figure 12 in the appendix to show a breakdown of the reward by unique graph topology on the TF runtime test set (note that as per section A.1.2 we created 99 additional copies of each graph and randomly modified the tensor sizes). We see that for many graphs, the effect of REGAL (most often, the improvement from REGAL) is consistent within the family. This suggests that REGAL is identifying patterns that are specific to an architecture.
>
> > - It seems the method and assumptions about graphs or training data are pretty coupled with TensorFlow and graph-mode execution, how could the method be generalized to other ML frameworks (e.g. frameworks with eager execution)
>
> The methods and its assumptions are indeed tied to static scheduling; however, it’s not accurate to say that they are coupled to TensorFlow; they apply to any optimizing static compiler for neural network computation graphs. Such compilers include Glow, MLIR, TVM, and XLA. XLA, for example, can be used from TensorFlow, PyTorch, Jax, and Flux/Julia.
>
> For a pure eager-mode setting, our methods do not apply and would need to be substantially redesigned. The work of Mao et al. (2019) may be interesting in this regard, in that they apply learning to an on-line scheduling problem where both a schedule and mapping onto hardware must be decided as new jobs arrive to a data processing cluster.

---

> > ### Author Response · Authors · 2019-11-15
> > **Continuation**
> >
> > > - Could the authors clarify why the two methods mentioned in “Learning to directly predict a solution” has quadratic complexity w.r.t. # of nodes and whereas REGEL is linear?
> >
> > Let n be the number of nodes in the input graph for which placement and scheduling decisions need to be predicted. Predicting the decisions with an autoregressive model will need O(n) steps, where each step involves performing inference on the graph neural network. Since a single inference pass on the GNN has at least O(n) cost, the total prediction cost scales as O(n^2). We also experimented with a non-autoregressive approach for predicting the decisions that has O(n) total cost, but the results were significantly worse. REGAL performs a single inference pass on the GNN, so it has O(n) cost.
> >
> > > - Confusion on Figure 4(b): Could some more critical statistics about the graphs in the training/test dataset be reported? e.g. what’s the average depth of the training graphs? When there are 32 MP layers a node’s feature will be passed across its 32-hop neighborhood, which seems surprising as it is common to observe GNN starts degenerating with increased depth …
> >
> > We added Figure 6 in the appendix to show the distribution of the diameters of graphs in our dataset.
> >
> > We do observe a plateau of GNN performance with increased depth (as reported in Figure 4(b)), but no significant drop with large depth.  In principle, even when the number of layers is larger than the graph diameter, the GNN can still use the additional layers to do more computation, which can be helpful for making predictions.  [Selsam et al. (2019)] (https://arxiv.org/abs/1802.03685 ) shows an extreme example of this where 1000 message passing layers were used in GNNs to make predictions on graphs with much less number of nodes, and improved performance was reported with increased number of message passing layers even up to 1000.  Training GNNs with large depth may be more challenging than training shallower GNNs, but various techniques can be applied to make this easier, e.g. adding GRU / LSTM-style gating or residual connections. Overall we did not experience the degeneration at 32 message passing layers that the reviewer suspected.

---

### Official Review · AnonReviewer2 · 2019-10-29
**Official Blind Review #2**

**Rating:** 8

**Review:**

In this paper, the authors proposed a framework to generating a task scheduling for a compiler to reduce the execution cost of neural networks. A computation graph is first fed into a GNN to produce a beta distribution, which is then fed into the BRKGA algorithm to yield the encoded solutions. The motivation is interesting, and the proposed method is technically reasonable. The details are also included in the appendix. To improve the quality, the following concerns may be considered:

1. The detailed explanations of o_a(G) and o_s(G) should be included.

2. How were the attribute vectors  x_v and x_e defined in your experiments?

3. The baseline (GP+DFS) may not be strong enough, since it is designed to reduce the communication cost. With the information of the input size and time complexity of ops, a better greedy algorithm can be designed. Moreover, the performance of Local Search and BRKGA 5K are similar, and REGAL is just slightly better than BRKGA 5K. Hence, the improvement over the best efficient greedy algorithm seems small.

Overall, the studied topic is interesting, and this paper is also intriguing.


**Experience Assessment:**

I have read many papers in this area.

**Review Assessment: Checking Correctness Of Derivations And Theory:**

I carefully checked the derivations and theory.

**Review Assessment: Checking Correctness Of Experiments:**

I carefully checked the experiments.

**Review Assessment: Thoroughness In Paper Reading:**

I read the paper at least twice and used my best judgement in assessing the paper.

---

> ### Author Response · Authors · 2019-11-15
> **Response to "Official Blind Review #2"**
>
> Thank you for the review and the interesting questions!
>
> > 1. The detailed explanations of o_a(G) and o_s(G) should be included.
>
> o_a(G) and o_s(G) are defined as the objective value of the best solution for graph G found by BRKGA using, respective, 1) the mutant sampling distributions predicted by the GNN, and 2) uniform distributions (i.e., as is done in standard BRKGA).
>
> Multiple reviewers have requested additional details on how BRKGA works, so we have added a self-contained description of the meta-heuristic algorithm to Section 3.2.
>
> > 2. How were the attribute vectors  x_v and x_e defined in your experiments?
>
> The specific node features x_v and edge features x_e have are described in section A.2 of the appendix. We have expanded on this description.
>
> > 3. The baseline (GP+DFS) may not be strong enough, since it is designed to reduce the communication cost. With the information of the input size and time complexity of ops, a better greedy algorithm can be designed. Moreover, the performance of Local Search and BRKGA 5K are similar, and REGAL is just slightly better than BRKGA 5K. Hence, the improvement over the best efficient greedy algorithm seems small.
>
> We have acknowledged in Section 5.3 that GP+DFS is a weak baseline. We compare with it because a similar GP approach is used by XLA for model parallelism and by Mirhoseini et al. (ICML ‘17) as a baseline.
>
> If we understand your suggestion about a greedy algorithm, this would be one that sequentially decides which task to run next and on which device. One issue with this approach is that it could get stuck with no feasible moves to make due to the memory constraints. It would nevertheless be possible to try this (see, e.g., https://arxiv.org/abs/1711.01912 which we recently discovered), although we expect that Tuned BRKGA would provide higher-quality solutions.
>
> Also, the hyperparameters for local search were tuned using grid search the same way as Tuned BRKGA, so it should be compared to Tuned BRKGA rather than BRKGA 5K. The gap in percent improvement between local search and Tuned BRKGA is larger than the gap between local search and BRKGA for the TF Runtime and Synthetic Runtime test sets (Table 1).
>
> In our opinion, the improvements are not small, they have to be judged with respect to how difficult it is to obtain these improvements - see room for improvement (Table 1; “Gap from best known”), effort required to gain the same improvement for BRKGA (Fig 3) and absolute improvements (Fig 2).
>
> > Overall, the studied topic is interesting, and this paper is also intriguing.
>
> Thanks!

---

### Decision · Program_Chairs · 2019-12-19

**Decision:**

Accept (Poster)

**Comment:**

The submission presents an approach that leverages machine learning to optimize the placement and scheduling of computation graphs (such as TensorFlow graphs) by a compiler. The work is interesting and well-executed. All reviewers recommend accepting the paper.